# An Optimized CNN Model for Engagement Recognition in an E-Learning Environment

**Yan Hu** *⭐ , **Zeting Jiang** † **and Kaicheng Zhu** †

Department of Computer Science, Blekinge Institute of Technology, 371 79 Karlskrona, Sweden
* Correspondence: yan.hu@bth.se; Tel.: +46-455-385829
† These authors contributed equally to this work.

**Abstract:** In the wake of the restrictions imposed on social interactions due to the COVID-19 pandemic, traditional classroom education was replaced by distance education in many universities. Under the changed circumstances, students are required to learn more independently. The challenge for teachers has been to duly ascertain students' learning efficiency and engagement during online lectures. This paper proposes an optimized lightweight convolutional neural network (CNN) model for engagement recognition within a distance-learning setup through facial expressions. The ShuffleNet v2 architecture was selected, as this model can easily adapt to mobile platforms and deliver outstanding performance compared to other lightweight models. The proposed model was trained, tested, evaluated and compared with other CNN models. The results of our experiment showed that an optimized model based on the ShuffleNet v2 architecture with a change of activation function and the introduction of an attention mechanism provides the best performance concerning engagement recognition. Further, our proposed model outperforms many existing works in engagement recognition on the same database. Finally, this model is suitable for student engagement recognition for distance learning on mobile platforms.

**Keywords:** CNN; ShuffleNet v2; engagement recognition; E-learning

## 1. Introduction

The pandemic that has gripped the world for more than two years has significantly impacted the education sector. Online video-based teaching that was introduced during the pandemic has become more and more common. The influence of online education on learning efficiency has become an important question worth exploring, because many real-life conditions restrict online teaching when compared with traditional face-to-face education. Therefore, to understand the effects of online education on learning in detail, detecting and obtaining learners' engagement is quite essential for subsequent studies. Since learner engagement has always been a key topic in the field of education, some studies have indicated that learner engagement can be improved through appropriate instructional interventions, good study design and instant feedback [1]. To measure the engagement of learners, the human face is one of the starting points.

Facial expression are a very common, natural and universal way for humans to express their emotions. Earlier face-recognition methods use a face dataset to recognize one or more faces in a scene through static images. With improvements in camera performance and cost reduction in this technology in recent years, face recognition is widely used in games, mobile payments, video surveillance and many other areas. Consequently, a large amount of video data is generated, and the demand for processing faces in the video format is on the rise.

At the end of the last century and the beginning of this century, mainly traditional algorithms and machine learning algorithms were used for facial recognition. Common examples are Eigenfaces [2], Fisherfaces [3], Bayesian face [4], support vector machine

(SVM) [5], Boosting [6] and Metaface [7]. However, facial recognition systems that are based on these algorithms work well only when they operate in a relatively controlled environment [8]. As more variables are introduced in the video streaming data, such as resolution, motion blur and illumination, the technical challenges increase.

Around 2010, thanks to the rapid development of GPU performance and the fast growth in the number of real-world datasets, researchers gradually began to use deep learning methods for facial recognition. The most classic and popular example of this is the use of convolutional neural networks (CNNs) [9]. Other instances are long short-term memory (LSTM) [10], recurrent neural networks (RNNs) [11], 3D convolutions (C3Ds) [12] and so on. The results show that deep learning algorithms have achieved better recognition accuracy and greatly surpassed methods based on traditional machine learning [13]. Deep learning is a machine-learning technique used to build and simulate the ways the human brain analyzes, learns and interprets data. Deep learning uses a multi-layer network called a neural network, which is inspired by the human brain. It iteratively analyzes the data layer-by-layer and continuously extracts features to obtain the essential characteristics of the original data. When compared to traditional classification methods, deep learning shows better results solving classification problems. The most-notable applications of deep learning are computer vision and natural language processing [14,15].

In recent years, there have been many novel neural network structural models applied in various fields of computer vision [16,17]. However, there are limited studies and applications of engagement recognition based on videos in an e-learning environment. Sometimes, students prefer distance education to be carried out on a mobile terminal or a device with limited memory. Therefore, a real-time engagement-recognition model needs to consider speed and delay in particular. Although a typical neural network structure such as VGGNet [18] can have better performance of engagement recognition, it brings efficiency problems.

This study proposes a lightweight CNN model with acceptable accuracy to detect student engagement in an online-education environment. This model is lightweight without affecting its performance. To achieve this objective, we combine the attention mechanism and other methods to improve the accuracy of this model. The research questions for this study are as follows:

- **RQ 1**: Which CNN-based models are effective in recognizing student engagement in an E-learning environment?
- **RQ 2**: Which methods can optimize a CNN-based engagement-recognition model to adapt it to mobile devices?
- **RQ 3**: How does the optimized model perform compared with other selected models in recognizing student engagement in an E-learning environment?

The overall structure of this paper is as follows. In Section 2, the basic concepts and different models of CNN are introduced. The related work in applying deep learning algorithms in engagement recognition is listed in Section 3. Our improved model and the performance evaluation experiment design are described in Section 4. In Section 5, performance comparisons of four CNN-based models are presented, and the accuracy of the proposed model is also compared with the existing models using the same dataset. Section 6 analyzes and discusses our findings. Finally, Section 7 concludes our study and proposes some future work in this area.

## 2. Background

Convolutional neural networks (CNNs) are a class of artificial neural networks. Their artificial neurons can react to the surrounding units in the coverage area. They are usually used to process data with a grid pattern, such as images [19]. CNNs are similar to artificial neural networks, as they are composed of neurons with learnable weights and bias constants. Furthermore, CNNs are a feedforward artificial neural network [20], which allows the encoding of specific properties into the network structure and the introducing of weight-sharing mechanisms, making the feedforward function more efficient by reducing a large number of parameters. In addition, the down-sampling operation in CNNs can effectively increase the receptive field of the network, which helps to ensure the transla-

tion invariance of the image so that the network can have stronger feature extraction and characterization capabilities [17,21,22].

### 2.1. Activation Function

An activation function is meant to transfer the activated information to the next layer when activating a certain part of the neurons in the neural network [23]. As the distribution of data is mostly nonlinear and the calculation of the general neural network is linear, the activation function introduces nonlinearity into the neural network and strengthens the network's learning ability. Thus, the biggest feature of the activation function is nonlinearity. The activation function also has the characteristics of differentiability and monotonicity.

Several commonly used activation functions are Sigmoid, Tanh, ReLU, PReLU and Leaky ReLU. As an activation function, Sigmoid has the advantages of smoothness and easy derivation. Tanh results from downward translation and scaling of Sigmoid. When compared with Sigmoid and Tanh, the ReLU function abandons complex calculations and improves the calculation speed. The problem of vanishing gradient is also solved. Its convergence speed is faster than Sigmoid and Tanh. Further, by increasing the nonlinear mapping between the neural network layers, over-fitting can be effectively avoided [24]. However, ReLU forces the output of the part $x < 0$ to be set to 0; that is, it masks the feature, which may cause the model to fail to learn effective features. In this case, if the learning rate is set too large, it may cause most of the neurons in the network to be in 'dead' status. The Leaky ReLU function is used to solve this "death" problem.

### 2.2. Typical and Lightweight CNN Architecture

The most practical way to obtain a high-quality model is to increase the depth (number of layers) or width (number of layers or neurons) of the model [13,25–27]. Nowadays, the design of the model is mainly divided into four types. The first design type involves building a deeper network such as ResNet [25]; the second design type involves building a wider network such as Inception [26]; the third design type combines the characteristics of the first two design types to make a deeper and wider network with better performance [28]; and the fourth type is a lightweight model designed for mobile devices, such as MobileNet and ShuffleNet [27,29]. In distance learning, the end devices used to receive online video lectures are not limited to personal computers. Mobile devices are also widely used by students. Therefore, we would like to choose a lightweight CNN model in our study.

#### ShuffleNet

ShuffleNet v1 [27] is a lightweight convolutional neural network for mobile devices that was proposed by face++ at the end of 2017. The innovation of the network lies in the use of pointwise group convolution and channel shuffle to ensure the accuracy of the network while greatly reducing the required computing resources. Existing advanced basic architectures such as Inception and ResNet are less efficient in small network models because a large number of $1 \times 1 \times 1$ convolutions consume a lot of computing resources. ShuffletNet proposes pointwise group convolution to help reduce computational complexity. However, the use of pointwise group convolution will create a large amount of calculations, so on this basis, the model proposes channel shuffle to help information flow. In Figure 1, GConv stands for group convolution: (a) Two connected group convolutional layers with the same number of groups. Each output channel is only related to the input channels in the group. (b) When GConv2 gets data from different groups after GConv1, the input and output channels are completely related; (c) Using the channel rearrangement mechanism to achieve (b).

The efficient architecture of ShuffleNet is built based on these two technologies. When compared with other advanced models, for a given computational complexity budget, ShuffleNet allows the use of more feature-mapping channels, which helps to encode

smaller network information. Figure 2a,b shows the basic unit of ShuffleNet v1, which is improved based on a residual unit.

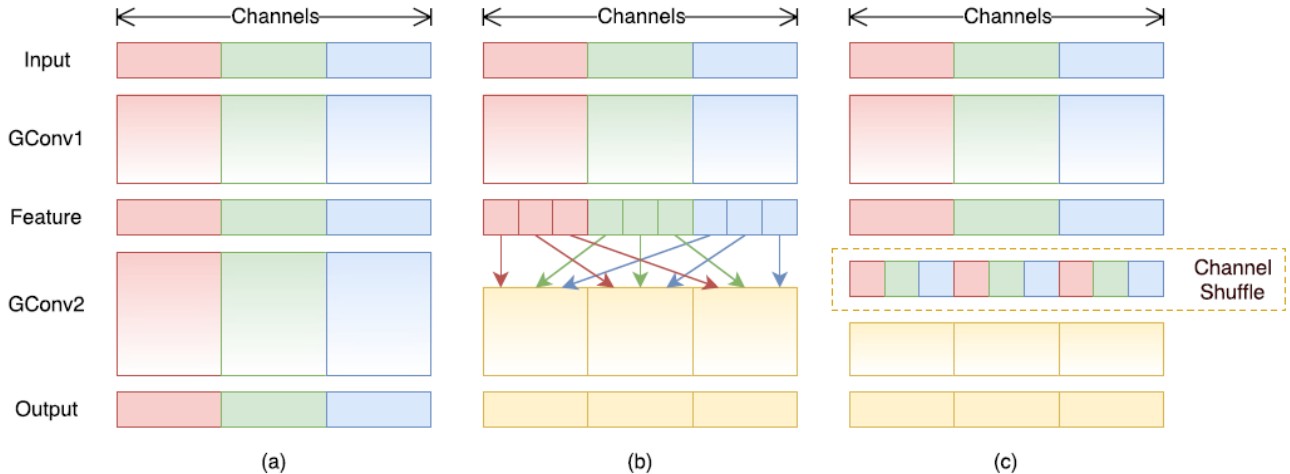

**Figure 1.** Channel shuffle [27].

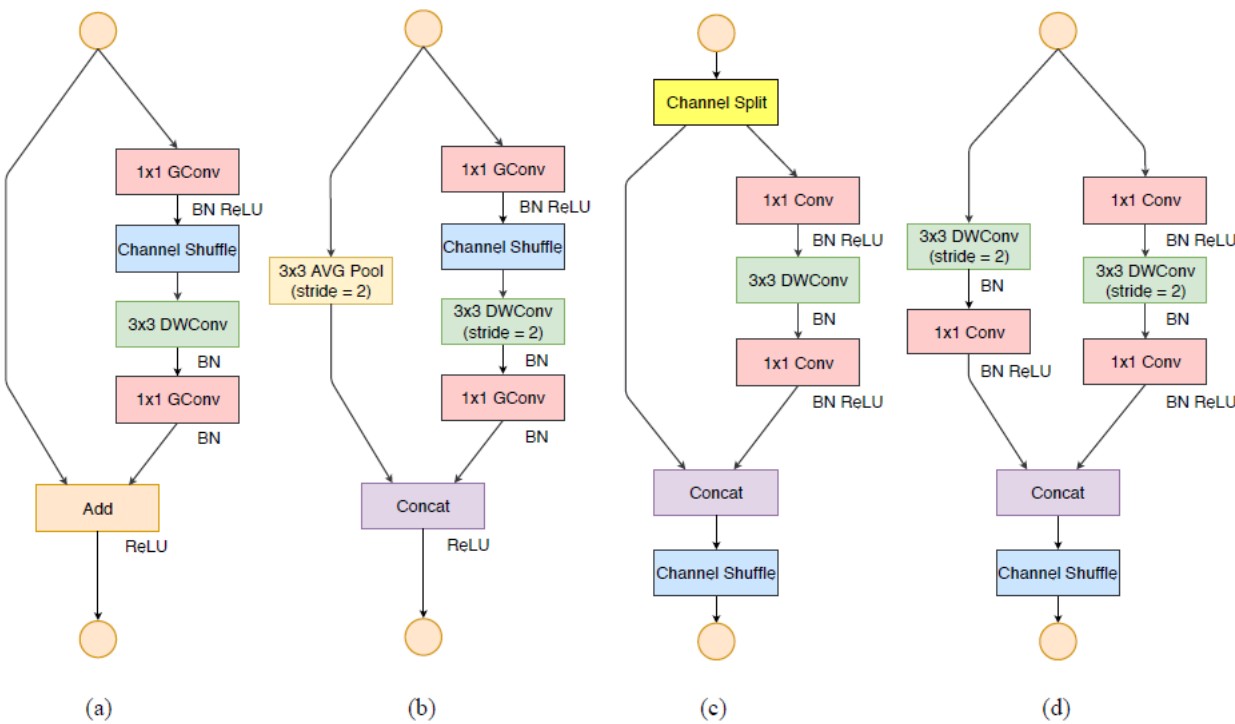

**Figure 2.** ShuffleNet units [30].

Ma, Zhang, Zheng and Sun [30] presented a state-of-the-art architecture of deep convolutional neural networks and named it ShuffleNet v2. The whole study was performed on two universal platforms: a single GPU and a mobile ARM. In their controlled experiments, they put forward two basic principles and four guidelines to guide network architecture design. Then, they compared and analyzed the runtime performance with MobileNet v2 [31], ShuffleNet v1 and other neural networks. It was found that ShuffleNet v2 is about 3.5% more accurate than the other two networks and achieved the best trade-off between speed and accuracy. The four guidelines proposed by ShuffleNet v2 for network structure optimization are the following:

1. When the input feature matrix of the convolutional layer and the output feature matrix channel are equal, the memory access cost is the smallest.
2. When the GConv groups increase (while keeping FLOPs unchanged), the MAC will also increase.
3. The higher the fragmentation of the network design, the slower the speed, which means more branches; hence, the speed is slower.
4. Element-wise, the impact of operations (such as ReLU, etc.) cannot be affected.

Figure 2 shows the basic units of ShuffleNet v1 and ShuffleNet v2; (a) and (b) are two different block structures of ShuffleNet v1. The difference between the two versions is that v2 reduces the feature-map size, which is similar to that in ResNet. The functions of the two blocks of a stage are similar. Similarly, (c) and (d) are two different block structures of ShuffleNet v2.

## 3. Related Work

Grafsgaard et al. [32] used a computer expression-recognition toolbox to recognize facial expressions to establish a predictive model based on the micro-expressions of the observed person, such as raised outer brow, tightened lips and so on. Sharma et al. [33] presented a system to detect the engagement level of students based on images. They applied the Viola–Jones algorithm first to detect the student's face. Then, the eye region was fed into a CNN as a binary classifier to predict the student's attention state in either "distracted" or "focused" categories based on the movement of the head and eyes. After this, they created another CNN model that recognizes the dominant emotion expressed by the student's face at each moment. Finally, they used a questionnaire to collect the influence of different expressions on engagement and assigned different weights to judge the influence of facial expressions on engagement. Thus, the proposed system predicts student engagement by calculating an engagement index using the confidence level of dominant emotion and emotion weights.

The authors in the above papers usually used the weight of the facial expressions corresponding to the degree of engagement to predict the engagement. Nevertheless, Nezami et al. [18] created their own engagement dataset. They quantified and characterized engagement using facial expressions extracted from images. They first used a CNN during basic facial expression recognition. Then, they applied the generated model, similar to the VGG-B architecture, to initialize the engagement-recognition model. During the recognition stage, they designed two dimensions to measure: behavioral and emotional. In the dimension of emotion, they defined three emotions to describe engagement, Satisfied to describe students' enthusiasm for learning, and Confused and Bored to describe the negative states of students. This engagement model generated based on the above conditions achieved considerable improvement.

It can be seen that the engagement recognition task is similar to the facial expression recognition task. Therefore, engagement recognition can be divided into static engagement recognition and dynamic engagement recognition. Static engagement recognition recognizes a person's engagement through a single picture, while dynamic engagement recognition recognizes a person's engagement through a video/picture sequence. Unlike the above articles, Whitehill et al. [34] annotated a 10-s video dataset into four levels of engagement and applied a linear support vector machine as well as the Gabor features to recognize student engagement. Moncaresi et al. [35] used facial features and heart rate features to detect engagement. They used self-reported data collected from students during and after the task to annotate their video dataset into two categories: engagement and disengagement. The machine-learning tool WEKA was used to classify the categories of engagement. Gupta et al. [36] proposed and introduced DAiSEE, a multi-label video classification dataset. This dataset uses MOOCs as the e-learning environment. This article mainly started from the usage of the video dataset, and the authors also applied some classic network structures of CNN to analyze the practicality of the dataset. In 2022, Jagadesh and Baranidharan [37] introduced their own real-time online learning videos dataset.

Each video from this dataset consists of 50 frames, and a total of 100 videos were used in their experiment. Further, they proposed a hybrid model based on both CNN and RNN models to improve performance.

As one of the few open datasets consisting of videos, different engagement-recognition models were proposed by using the DAiSEE dataset, such as different long short-term memory (LSTM) models [38] and neural Turing machine [39]. Liao et al. [40] presented a deep facial spatiotemporal network (DFSTN) for engagement detection. This model contains the following two modules: a pre-trained SE-ResNet-50 (SENet) for extracting facial spatial features and an LSTM network with global attention (GALN) to generate a hidden attention state. Further, different hybrid neural network architectures created by combining different networks have also been proposed, such as a deep engagement recognition network (DERN) [41], a residual network (ResNet), a temporal convolutional network (TCN) (ResNet+TCN) [42] and so on.

As for video and image recognition and classification, there are many typical neural network models. For example, inception_v4 [28] and ResNet [25] proposed in recent years have greatly improved the accuracy of classification and recognition models. However, lightweight models can better solve the efficiency problem for mobile or embedded devices. A study by Boulanger et al. [43] found that a lightweight CNN model, SqueezeNet, which has 50x fewer paraments, does not reach the same accuracy and effectiveness as traditional CNN models for detection of engagement for online learners. The Google DeepMind team [44] introduced an attention mechanism into a recurrent neural network model for image classification. The attention mechanism was borrowed from human attention by quickly scanning the global image to obtain the target area to focus on and then devoting more resources to this area to obtain more detailed information. The model selects the next position that should be noticed at each step based on the past information and the requirements of a given situation rather than processing the whole image at one time, thus improving performance and saving a lot of time and hardware resources. To improve the expressive ability of the network, Hu et al. [45] focused on the channel relationship, and they introduced a new architectural unit SE block. The SE block can model the interdependence relationship between channels of convolutional features to improve the network's representation ability. To achieve this goal, they proposed a mechanism that allows the network to perform feature recalibration. Through this mechanism, the network can learn to use global information to selectively emphasize features with large amounts of information and suppress the less-useful features. In this way, these SE modules are stacked together to build an SENet architecture with good generalization ability on challenging datasets. Shen et al. [46] proposed a novel lightweight assessment system for learning engagement recognition that introduced SE modules. However, the performance testing of the experiment from their paper was only based on static images.

Based on all the above papers, CNN has achieved breakthrough results in computer vision, which makes it very suitable for solving the task of engagement recognition. Obviously, there are few open-source engagement-recognition datasets for engagement recognition tasks, so there has not been much research in this area. The research and application of the practicality of the model are even rarer. Therefore, this paper aims to develop a high-efficiency model using a lightweight CNN to recognize engagement, to propose some optimization measures and to compare performance with existing state-of-the-art models.

## 4. Method

### 4.1. Proposed Model Architecture

As an extremely efficient CNN architecture, ShuffleNet v2 can be applied explicitly to mobile devices with limited computing power [30]. The calculation amount is significantly reduced with similar accuracy when compared with existing advanced models. Among different lightweight CNN models, ShuffleNet v2 has also verified its better performance on many direct metrics, such as speed, latency and memory access cost, rather than

only approximate factors such float-point operations (FLOPs) [30]. Therefore, we considered optimizing a ShuffleNet-based model to improve its accuracy without affecting other performance parameters as much as possible. From the perspective of model practicability, this paper chooses a ShuffleNet v2 0.5x structure to minimize computational complexity. The computational complexity of ShuffleNet v2 with different sizes is shown in Table 1.

**Table 1.** Computational complexity of ShuffleNet v2 with different sizes [30].

| Network Size | Computational Complexity(FLOPs) |
|---|---|
| ShuffleNet v2 0.5x | 41 M |
| ShuffleNet v2 1.0x | 146 M |
| ShuffleNet v2 1.5x | 299 M |
| ShuffleNet v2 2.0x | 591 M |

According to the activation function mentioned in the background section, the Leaky ReLU function does not only avoid the problem of gradient disappearance, but it also solves the problem of neuron "death" phenomenon, which can effectively improve the accuracy of the model [47]. Therefore, this study conducts a pre-experiment to compare different activation functions trained on the selected dataset. Based on the pre-experimental results, the function with the best accuracy is selected as the activation function of the engagement-recognition model.

Replacing the activation function with better accuracy is the first measure to optimize the engagement-recognition model, and the second to consider the attention mechanism's application. It was mentioned in the related work that the attention mechanism has been widely used in computer vision. Moreover, the attention mechanism can improve accuracy with little cost in terms of calculation requirements. Our attention mechanism module adopts the SE block mentioned in the Related Work Section. The SENet attention mechanism has two main operations: squeeze and excitation. Squeeze refers to compressing the spatial information of features through GAP, and compressing the original $c \times h \times w$ dimension information to $c \times 1 \times 1$. Excitation uses two fully connected layers. The first reduces the dimension, and $c \times 1 \times 1$. The dimension is reduced to $c/r \times 1 \times 1$ (with ReLU activation), and the second fc layer remaps the features back to $c \times 1 \times 1$ (without ReLU activation); then, after sigmoid, the weight coefficients of each channel are obtained. Then, we multiply the weight coefficient with the original feature to get a new feature. Therefore, the network structure after the embedded attention module based on the ShuffleNet v2 0.5x architecture is shown in Table 2.

**Table 2.** Engagement-recognition model structure.

| Output Size | ShuffleNet v2 + Attention Module | Repeat |
|---|---|---|
| 112 × 112 | Conv 3 × 3, 24, stride 2 | 1 |
| 56 × 56 | MaxPool 3 × 3, stride 2 | 1 |
| 28 × 28 | Conv 1 × 1, 48<br>DWConv 3 × 3, 48<br>Conv 1 × 1, 48<br>Attention Module<br>Shuffle | 4 |
| 14 × 14 | Conv 1 × 1, 96<br>DWConv 3 × 3, 96<br>Conv 1 × 1, 96<br>Attention Module<br>Shuffle | 8 |

**Table 2.** *Cont.*

| Output Size | ShuffleNet v2 + Attention Module | Repeat |
| --- | --- | --- |
| 7 × 7 | Conv 1 × 1, 192<br>DWConv 3 × 3, 192<br>Conv 1 × 1, 192<br>Attention Module<br>Shuffle | 1 |
| 7 × 7 | Conv 1 × 1, 1024 | 1 |
| 1 × 1 | GAP + fc + softmax | 1 |

### 4.2. Experimental Setup

After our experimental code is implemented locally, our proposed model and the comparison models are trained and tested on a GPU cloud server with the configuration shown in Table 3. In our study, all model training and verification uses CUDA 11.0 and a CuDNN 8.0 accelerated computing library in Python 3.6 and the PyTorch 1.9.0 environment.

**Table 3.** GPU cloud server configuration.

| CPU | Intel(R) Xeon(R) Gold 6142 CPU @ 2.60 GHz |
| --- | --- |
| GPU | NVIDIA GeForce RTX 3080 |
| Number of CPU cores | 8 |
| RAM | 10 GB |
| Floating-point operations per second | Half-precision 29.77 TFLOPS/Single-precision 29.77 TFLOPS |

### 4.3. Engagement Recognition Dataset

Due to the impact of COVID-19, it has become impossible to collect data about student engagement in the classroom or to imitate the different forms of online education environments. It is challenging to create a unique dataset. There are fewer video datasets in this specific environment for E-learning. In this case, DAiSEE [36] is the only open-source dataset that can be used, as mentioned in the Related Works section. DAiSEE contains 9068 videos captured from 112 students. They are all Asian students, comprising 32 females and 80 males aged between 18 and 30 years. The subjects were required to watch a 20 min online educational video. There are six different data-collection locations, such as dorm rooms, crowded lab spaces and libraries, which offer a good simulation of real-world cases. The dataset contains the following four emotional states related to user engagement: engagement, frustration, confusion and boredom. Further, there are the following four labels under each engagement category: very low, low, high and very high. This paper mainly focuses on the state of engagement. The video length, frame-rate and resolution are 10 s, 30 frames per second (fps) and 1920 × 1080 pixels.

### 4.4. Model Training

Before training the model, each video is down-sampled to obtain $3 \times 224 \times 224$ (C × H × W) tensors. After that, three CNN pre-training models are downloaded. The architectures are Inception-v3, ResNet50 and ShuffleNet v2 0.5x. These three models and our optimized engagement-recognition model are trained on the ImageNet dataset. The ImageNet dataset is a large visual database designed for use in visual object recognition [13]. We use stochastic gradient descent (SGD) for these models for parameter optimization. The learning rate and batch size for Inception v3 are 0.001 and 64, respectively. For ResNet50, the learning rate and batch size are 0.01 and 64, respectively. For Shufflenet v2 0.5x baseline and our optimized engagement-recognition model, the learning rate and batch size are 0.001 and 64, respectively. After pre-training is completed,

the network parameters of these models must be fine-tuned and corrected with training samples to improve system classification.

### 4.5. Model Evaluation

After model training is completed, to evaluate the performance, strengths and weaknesses of our proposed engagement-recognition model, metrics such as accuracy, F-measure (F-score) and FLOPs are often used. The definitions and calculation formulas of some indicators are listed hereafter.

#### 4.5.1. Accuracy

Accuracy is defined as the number of correct predictions made by the model as a percentage of the total number of predictions.

$$Accuracy = \frac{Number \quad of \quad correct \quad predictions}{Total \quad number \quad of \quad predictions} \tag{1}$$

$$Accuracy = \frac{TP + TN}{TP + TN + FP + FN} \tag{2}$$

#### 4.5.2. F-Measure

F-measure is a common metric for measuring the accuracy of a method and is often used to determine the accuracy of an algorithm. Precision and Recall are often mentioned separately in recognition and detection-related algorithms, and the F-measure is a balanced reflection of the accuracy of the algorithm by considering both of these values.

Precision and Recall are defined as follows:

$$Precision = \frac{TP}{TP + FP} \tag{3}$$

$$Recall = \frac{TP}{TP + FN} \tag{4}$$

In the above two formulas, some basic concepts of metric usage can be defined as follows:

- **True Positive (TP):** Both the model-derived classification results and the actual classification results of the labels are engagement.
- **False Positive (FP):** The classification result obtained by the model is engagement, but the actual classification result of the label is not.
- **True Negative (TN):** Neither the model-derived classification results nor the actual classification results of the labels are engagement.
- **False Negative (FN):** The classification result obtained by the model is not engagement, but the actual classification result of the label is engagement.

Generally, higher accuracy is better, but precision and recall sometimes conflict, and the F-measure is more balanced, which also means that the F-measure is a better metric of performance.

The F-measure is defined as follows:

$$F_\beta = (1 + \beta)^2 \frac{Precision \times Recall}{\beta^2 (Precision + Recall)} \tag{5}$$

where $\beta$ is a user-defined parameter, and when $\beta = 1$, the F-measure is called F1-measure.

#### 4.5.3. FLOPs

Floating point operations (FLOPs), can be used to measure the complexity of an algorithm or model; the formula is as follows.

$$FLOPs = \left(2 \times C_i \times K^2 - 1\right) \times H \times W \times C_o \tag{6}$$

where $C_i$ is the input channel, $K$ is the kernel size, $H \times W$ is the output feature map size and $C_o$ is the output channel.

### 4.5.4. Recognition Speed

Because multiple CNN models are used, the detection speed for each frame in the video can also be used as a metric. The unit used for the detection speed parameter is milliseconds.

## 5. Results

### 5.1. Image Data Augmentation

In an effective deep learning model, the validation error must keep decreasing with the training error, and data augmentation is a powerful technique to reduce overfitting [48]. Figure 3 shows an example of our experiment using image augmentation techniques on an image. As clearly seen from this example, we randomly select a part of the image at a time and stretch, squeeze, rotate, distort and adjust the lighting on the selected part.

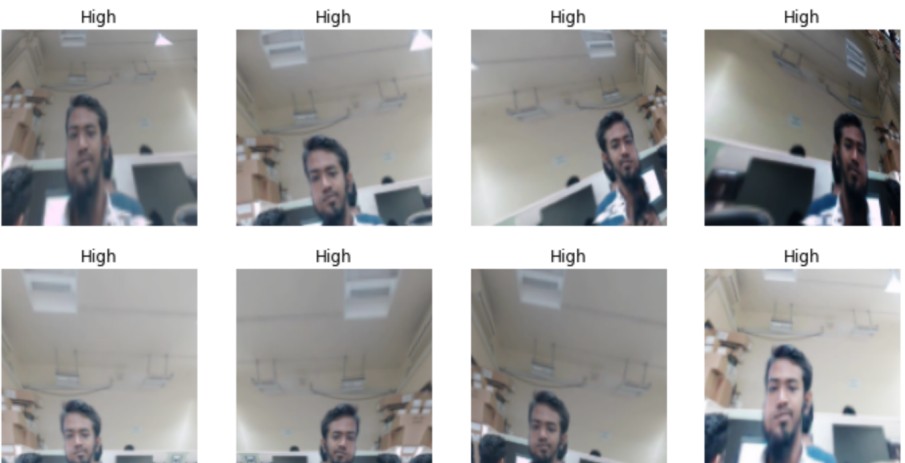

**Figure 3.** Image data augmentation examples.

### 5.2. Batch Examples

During the experiment, 64 images at a time were randomly selected for training. Figure 4 includes eight of those pictures. Each image in the example is implemented with image augmentation, and the engagement level is marked above each image.

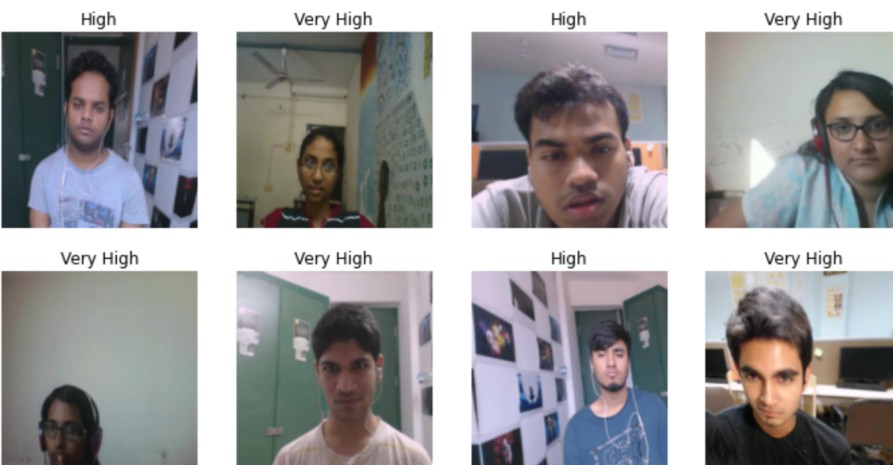

**Figure 4.** Batch examples.

### 5.3. Activation Function

As a pre-experiment, we verify the influence of different activation functions on model accuracy to choose the best activation function. According to the data presented in the Table 4, the respective accuracies of Sigmoid, Tanh, ReLU and Leaky ReLU are 54.5%, 53.9%, 57.7% and 58.6%. Among these activation functions, Leaky ReLU obtains the highest prediction, which is about 1% more accurate than the original activation function ReLU. Thus, our optimized ShuffleNet model uses the Leaky ReLU function as the activation function.

**Table 4.** Comparison of the accuracy of ShuffleNet v2 with different activation functions.

| Dataset | Sigmoid | Tanh | ReLU | Leaky ReLU |
|---------|---------|------|------|------------|
| DAiSEE | 0.5445 | 0.5392 | 0.5773 | 0.5862 |

### 5.4. Accuracy

According to the data presented in Figure 5, the respective accuracies of Inception v3, ResNet, ShuffleNet v2 and optimized ShuffleNet v2 models are 46.3%, 53.1%, 57.3% and 63.9%. Optimized ShuffleNet v2 has the best accuracy.

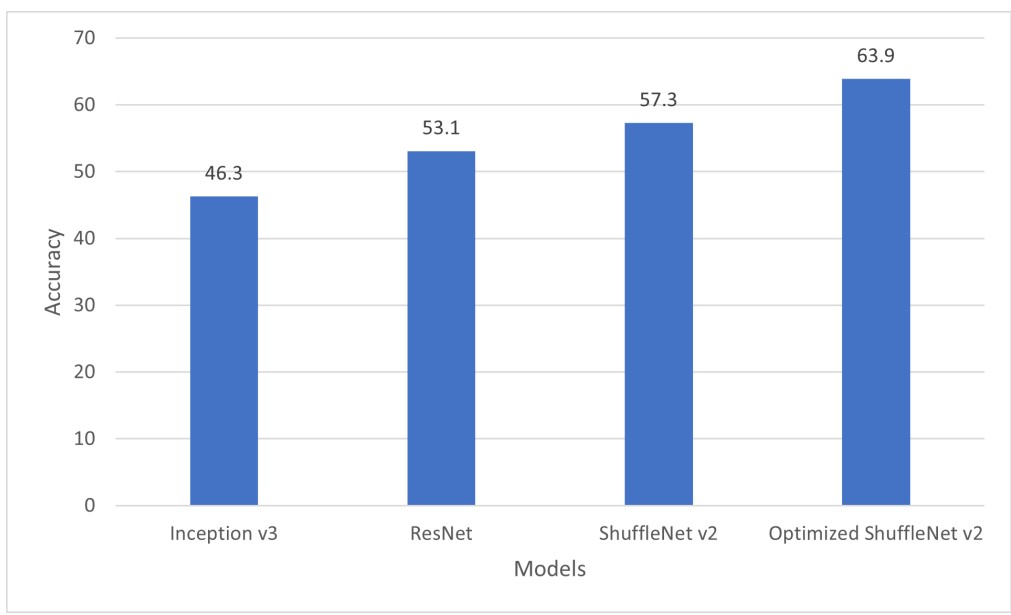

**Figure 5.** Classification accuracy of different models.

### 5.5. F-Measure

The precision of the model depends on the values of true positives and false positives, which are obtained by running the model on the test set.

According to the data presented in Figure 6, the respective precisions of Inception v3, ResNet, ShuffleNet v2 and optimized ShuffleNet v2 models are 0.620, 0.674, 0.683 and 0.783. ResNet's precision is about 0.05 better than Inception v3's precision. The precision of Shufflenet v2 with the change of activation function and the introduction of the attention mechanism is again about 0.1 higher than the original one.

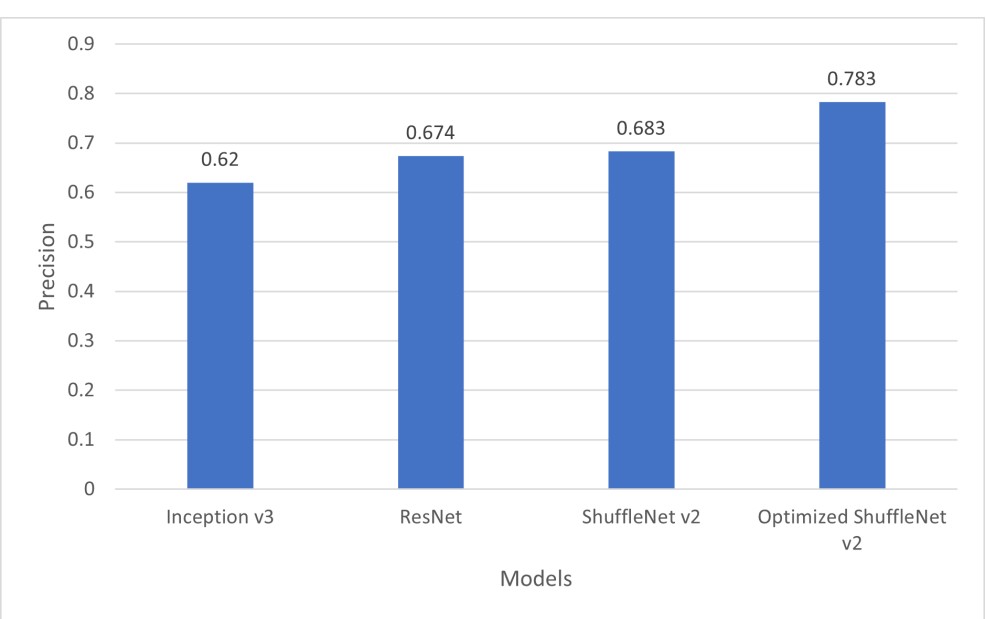

**Figure 6.** Classification precision of different models.

The recall of the model depends on the values of true positives and false negatives. According to the data presented in Figure 7, the respective recalls of Inception v3, ResNet, ShuffleNet v2 and optimized ShuffleNet v2 models are 0.534, 0.587, 0.623 and 0.692. The recall value of Inception v3 and ResNet is lower than that of ShuffleNet v2.

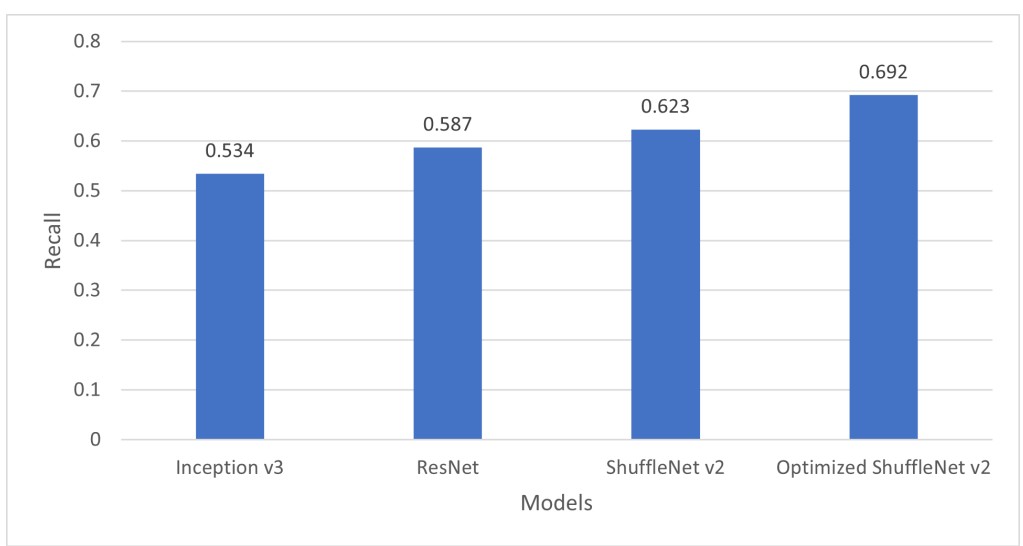

**Figure 7.** Classification recall of different models.

The F1-score is calculated based on the previously obtained precision and recall values. The F1-scores of Inception v3, ResNet, basic ShuffleNet v2 and optimized ShuffleNet v2 are 0.574, 0.627, 0.652 and 0.735, respectively, as shown in Figure 8. Optimized ShuffleNet v2 has the highest F1 score, while Inception v3 has the lowest F1-score.

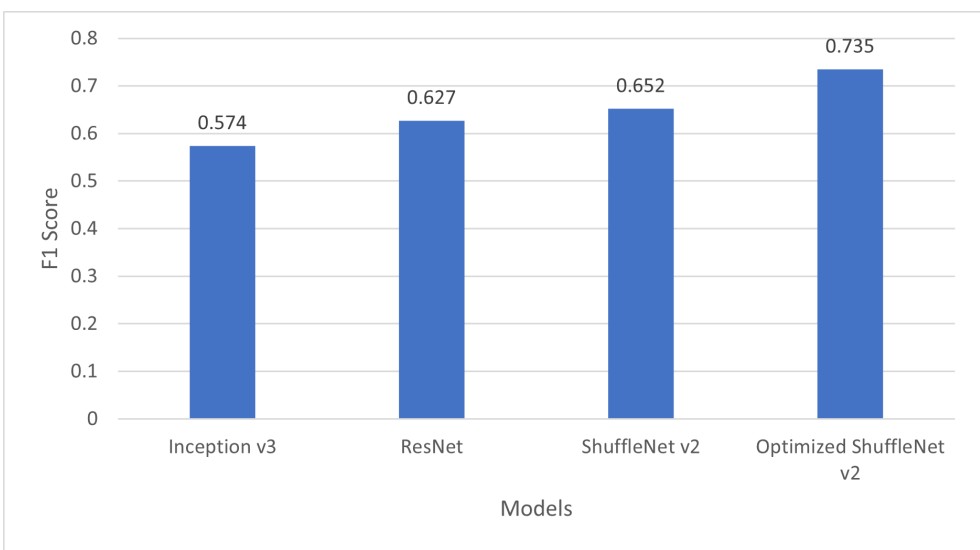

**Figure 8.** F1-score of different models.

### 5.6. FLOPs

As shown in Table 5, the FLOPs of Inception v3, ResNet, ShuffleNet v2 and optimized ShuffleNet v2 models are 140.26 M, 142.03 M, 43.69 M and 44.14 M, respectively. Among these models, the baseline ShuffleNet v2 has the lowest FLOPs, while ResNet has the highest FLOPs. For optimized ShuffleNet v2, its FLOPs are slightly higher than that of the baseline ShuffleNet v2 but significantly lower than that of Inception v3 and ResNet.

**Table 5.** FLOPs of different models.

| Model | Inception v3 | ResNet | ShuffleNet v2 0.5 | Optimized ShuffleNet v2 |
|---|---|---|---|---|
| FLOPs | 140.26 M | 142.03 M | 43.69 M | 44.14 M |

### 5.7. Speed

As shown in Table 6, the respective recognition speeds of Inception v3, ResNet, ShuffleNet v2 and optimized ShuffleNet v2 models are 458 ms, 514 ms, 317 ms and 362 ms. Among these models, baseline ShuffleNet v2 has the quickest recognition speed, while ResNet has the slowest recognition speed. The recognition speed of optimized ShuffleNet v2 is slightly less than that of the baseline ShuffleNet v2 but significantly more than Inception v3 and ResNet.

**Table 6.** Recognition speed of different models.

| Model | Inception v3 | ResNet | ShuffleNet v2 0.5 | Optimized ShuffleNet v2 |
|---|---|---|---|---|
| speed | 458 ms | 514 ms | 317 ms | 362 ms |

### 5.8. Comparison with Other Published Works

As we have described in our related work, there are several different models and techniques that have been proposed by others to detect student engagement with the same database that was used in our study. We took the reported accuracy from the following methods for comparison: convolutional 3D (C3D) [36], long-term recurrent convolutional network (LRCN) [36], DFSTN [40], C3D+TCN [42], DERN [41], neural Turing machine [39], ResNet+LSTM [42], ResNet+TCN [42] and (latent affective+behavioral+affect) features+TCN [49]. The results are shown in Table 7.

**Table 7.** Accuracy comparison with other published works.

| Models or Methods | Accuracy (%) |
| --- | --- |
| Convolutional 3D (C3D) [36] | 48.1 |
| Long-Term Recurrent Convolutional Network (LRCN) [36] | 57.9 |
| DFSTN [40] | 58.8 |
| C3D+TCN [42] | 59.9 |
| DERN [41] | 60.0 |
| Neural Turing Machine [39] | 61.3 |
| ResNet+LSTM [42] | 61.5 |
| ResNet+TCN [42] | 63.9 |
| (Latent Affective+Behavioral+Affect) features+TCN [49] | 63.3 |
| **Our Optimized ShuffleNet v2** | 63.9 |

From the table, it can be inferred that our proposed model outperforms almost all of the other recently published competitive methods using the DAiSEE database. Only ResNet + TCN reaches the same accuracy as our model.

## 6. Discussion

### 6.1. Answering the Research Questions

*RQ 1: Which CNN-based models are effective for recognizing student engagement in an E-learning environment?*

After our literature review, we decided to use four CNN models: Inception v3, ResNet, basic ShuffleNet v2 (which is a lightweight CNN model designed for mobile platforms), and an optimized ShuffleNet v2 model with an improved activation function and the introduction of an attention mechanism. These models are all able to perform engagement recognition according to our experiments, and ShuffleNet v2 was the main baseline model we used. Inception v3 and ResNet models were used for additional comparison.

*RQ 2: Which methods optimize CNN-based engagement recognition to adapt it to mobile devices?*

We suggest the use of attention mechanism SE block as a possible optimization method, with more details mentioned above. Further, introducing the activation function also optimized the model. Finally, using Leaky ReLU rather than ReLU achieves better results.

*RQ 3: How does the optimized model perform compared with other selected models in recognizing student engagement in an E-learning environment?*

We conducted an experiment to evaluate the recognition performance of these deep learning models. After all of the models were implemented and the data were processed, the models were trained on the training dataset. Then, the trained models were tested on the test dataset. The models' accuracy, precision, recall and final F1-scores were calculated. Optimized ShuffleNet v2 had the best performance among the deep learning models that were selected for this study. Furthermore, the accuracy of engagement recognition of our optimized ShufleNet v2 model was also superior compared to the other published methods and models using the same dataset.

### 6.2. Contributions

The main contribution of our study is the proposition of a novel, optimized, lightweight CNN model to recognize student engagement in an e-learning environment. Performance comparison of our model to other models and methods shows that the ShuffleNet v2 architecture reduces the complexity of the network, and the accuracy and recognition speed are better than those of currently existing architectures. The SE block applied to the Shuf-

fleNet architecture in our model also improves its accuracy. At the same time, it has little effect on the complexity and recognition speed of the model. In the proposed model, we used the Leaky ReLU function instead of the original ReLU function, as the activation function brings better accuracy, so replacing the activation function in our study optimized the model. Therefore, the activation function when subject to different tasks does impact the performance of the model. We believe our work points to the possibility of real-time engagement detection in distance learning on mobile devices.

## 7. Conclusions and Future Work

This paper proposes an optimized CNN model based on the lightweight architecture, ShuffleNet v2. We chose the ShuffleNet v2 0.5x structure to minimize computational complexity. We then introduced an attention mechanism for the model using SE blocks and improved the activation function. To evaluate and compare the performance, we selected three other CNN models: Inception v3, ResNet and ShuffleNet v2. Together with our proposed model, the selected models were then trained for engagement recognition. We used accuracy, F1-score, FLOPs and speed to measure the performance of these models. Our optimized ShuffleNet v2 model had the best performance based on the results of our experiment. Moreover, our model also achieved the highest accuracy when compared with the other published models or methods using the same dataset. As a lightweight framework, the optimized ShuffleNet v2 model is suitable for deployment on mobile platforms in E-learning environments.

In this paper, although our optimized ShuffleNet model with an attention mechanism reaches acceptable accuracy when compared with engagement-recognition models designed and implemented in other studies, we prioritize practicality. We had good performance concerning model complexity and recognition speed, but there is still room for improvement in terms of accuracy. For one, we found that the scale of the video dataset we used was still not large enough. Further, our model does not analyze timing-related information. Concerning the first issue, there are very few research studies and datasets in the area of engagement recognition. Even the DAiSEE dataset suffers from unbalanced sample distribution. There are far fewer samples with a low engagement level than with a high engagement level. This situation cannot be greatly improved with the current dataset. However, testing our proposed model with a larger and more-balanced sample could be a focus of future research work. Concerning the second issue, it must be noted that when optimizing ShuffleNet, we considered that increasing the analysis of timing information would inevitably affect the amount of calculation required for the model, which would undercut our goal of being lightweight. The expansion of the size of the dataset and the use of lightweight models that can analyze time-series information are other potential areas for future studies. Current engagement-recognition models are limited to using only CNNs. Future researchers can compare the performance other neural network structures or lightweight models. In future experiments in this area, researchers can also consider optimizing other variables, such as using a larger batch size, increasing the number of layers in the neural network or using a higher sampling rate for image augmentation for possibly better results. Applying our proposed model to detect engagement in real-life higher education E-learning environments is another main direction for future research work. Given that ShuffleNet v2 is a lightweight neural network, follow-up studies based on micro devices, such as tablets and mobile phones containing GPU chips, can be considered to explore the feasibility of building miniaturized engagement-detection devices.

**Author Contributions:** Conceptualization, Y.H., Z.J. and K.Z.; methodology, Y.H., Z.J. and K.Z.; software, Z.J. and K.Z.; validation, Z.J. and K.Z.; formal analysis, Z.J. and K.Z.; investigation, Y.H.; resources, Y.H., Z.J. and K.Z.; data curation, Y.H., Z.J. and K.Z.; writing—original draft preparation, Y.H., Z.J. and K.Z.; writing—review and editing, Y.H.; visualization, Z.J. and K.Z.; supervision, Y.H.; project administration, Y.H. All authors have read and agreed to the published version of the manuscript.

**Funding:** This research received no external funding.

**Institutional Review Board Statement:** Not applicable.

**Informed Consent Statement:** Not applicable.

**Data Availability Statement:** "DAiSEE Dataset" at https://paperswithcode.com/dataset/daisee, accessed on 29 June 2022.

**Conflicts of Interest:** The authors declare no conflict of interest.

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
