# Peer review of "An Optimized CNN Model for Engagement Recognition in an E-Learning Environment"

_applsci, doi:10.3390/app12168007_

Round 1

Reviewer 1 Report

The aim of the paper is not clear - pre-defining is needed. 

I can recommend to add research questions that should be commented in the study and conclusion.

References' numbers start from 40. They should be re-numbered from 1. The numbers of the sources are not sufficient - literature should be extended. 

Author Response

Dear reviewer, 

Thanks very much for your time to read the manuscript and provide your constructive comments for improvements. 

Here lists our response to all your comments:

  • The aim of the paper is not clear - pre-defining is needed. 

We have modified and added the aims in one separate paragraph, from line 63 to line 67.

  • I can recommend to add research questions that should be commented in the study and conclusion.

The research questions are added at the end of the Introduction section, starting from line 68. All the three research questions are answered in the begeinng of the Disucssion section, starting from line 422.

  • References' numbers start from 40. They should be re-numbered from 1. The numbers of sources are not sufficient - literature should be extended. 

References number now starts from 1, the Related work section is extended by more literatures, especially the works published from 2020 to 2022.

Thanks again for your time to read our revised version and this response letter to your comments.

Best Regards,

Yan Hu

Reviewer 2 Report

This paper really needs to be carefully edited by someone with english abilities similar to a native speaker. While the writing is overall reasonably clear, there are many small issues, such as typos, missing words, strange sentence constructions, etc…, that make the article hard to read and evaluate.  Bellow are some examples of these issues, but there are many more throughout the text:

“The proposed model are trained” -> “The proposed model is trained”

“provides the best recognition performance on the engagement 9 recognition.” -> provides the best recognition performance on the engagement recognition problem” ?

“the human face is the 24 one of the starting points.” -> “the human face is one of the starting points.”

“environment[47]. And more variables are” -> ???

“and the fast growth of dataset numbers from the real world” -> “and the fast growth of the number of real world datasets”

“ in Section 5. while the Section 6 analyse “ ->  “in Section 5, while the Section 6 analyses”

“Activation function is to transfer the activated information “ ->  “Activation function is to transfer the activated information “

“Element-wise The impact of operations “ -> “Element-wise the impact of operations “

“Then they based on the movement of the head and eyes” -> ???

“and uses 239 different activation functions are trained on the selected dataset…”

Please don’t start sentences by “And…”

The paper does a reasonable work of introducing the area and describing related work. The problem being tackled and the approach the authors followed to do it are also stated in a fairly comprehensible way. The authors proposed to optimize a ShuffleNet CNN to train a model capable of engagement recognition through facial expressions on a publicly available dataset. 

There are a few issues with the proposed approach that I would like to highlight. The authors state that “three CNN pre-training models were downloaded, the architectures are Inception-v3, ResNet50, ShuffleNet v2 0.5x, these models and our optimized engagement recognition model are trained on the ImageNet dataset”. I can understand that the pre-trained NN were previously trained on the generic ImageNet dataset, but why was the same done with the new model? Should it not only be trained with the new database?

Experimental results are a simple comparison of the four networks performance on the chosen dataset, and, while the results seem indeed better for the optimized shufflenet I’m not sure if the different results are not simply the consequence of inception and resnet being huge generic pre-trained networks that have no particular reason to perform well in this particular problem. The new model is never compared with other approaches that are competitive in this problem.

Overall, I’m not confident that the results presented in this paper are significant enough to justify publication as a journal article.

Author Response

Dear reviewer,

Thanks for taking the time to read the manuscript and provide your constructive comments for improvements.

Here lists our response to all your comments:

  • This paper really needs to be carefully edited by someone with english abilities similar to a native speaker. While the writing is overall reasonably clear, there are many small issues, such as typos, missing words, strange sentence constructions, etc…, that make the article hard to read and evaluate.  Bellow are some examples of these issues, but there are many more throughout the text:

“The proposed model are trained” -> “The proposed model is trained”

“provides the best recognition performance on the engagement 9 recognition.” -> provides the best recognition performance on the engagement recognition problem” ?

“the human face is the 24 one of the starting points.” -> “the human face is one of the starting points.”

“environment[47]. And more variables are” -> ???

“and the fast growth of dataset numbers from the real world” -> “and the fast growth of the number of real world datasets”

“ in Section 5. while the Section 6 analyse “ ->  “in Section 5, while the Section 6 analyses”

“Activation function is to transfer the activated information “ ->  “Activation function is to transfer the activated information “

“Element-wise The impact of operations “ -> “Element-wise the impact of operations “

“Then they based on the movement of the head and eyes” -> ???

“and uses 239 different activation functions are trained on the selected dataset…”

Please don’t start sentences by “And…”

Our response: The whole manuscript's English writing is now carefully checked and improved. All the points you mentioned above are fixed. 

  • The paper does a reasonable work of introducing the area and describing related work. The problem being tackled and the approach the authors followed to do it are also stated in a fairly comprehensible way. The authors proposed to optimize a ShuffleNet CNN to train a model capable of engagement recognition through facial expressions on a publicly available dataset. 
  • There are a few issues with the proposed approach that I would like to highlight. The authors state that “three CNN pre-training models were downloaded, the architectures are Inception-v3, ResNet50, ShuffleNet v2 0.5x, these models and our optimized engagement recognition model are trained on the ImageNet dataset”. I can understand that the pre-trained NN were previously trained on the generic ImageNet dataset, but why was the same done with the new model? Should it not only be trained with the new database?

The new training dataset we used for this study is not large enough, so we feel it's better to pre-trained the new model on the generic ImageNet dataset to improve the performance.

Experimental results are a simple comparison of the four networks performance on the chosen dataset, and, while the results seem indeed better for the optimized shufflenet I’m not sure if the different results are not simply the consequence of inception and resnet being huge generic pre-trained networks that have no particular reason to perform well in this particular problem. The new model is never compared with other approaches that are competitive in this problem.

Our response: We have added a new subsection (section 5.8, line 410) to compare the accuracy of our proposed model and the published models that used the same dataset. The results show that our model reaches the best accuracy. We also mentioned in future work (line 491-493) that in the future, we would try to compare our model with other network structures or other lightweight models.

  • Overall, I’m not confident that the results presented in this paper are significant enough to justify publication as a journal article.

Our response: In the revised version, we have added a new subsection called 6.3 Contributions to classify our study's significant contribution to the area.

Thanks again for your time to read our revised version and this response letter to your comments.

Best Regards,

Yan Hu

Reviewer 3 Report

In this paper is approached four deep learning algorithms based on based on the ShuffleNet architecture to determine the engagement of the students trough facial expressions analyzing images.

The paper is organized on 7 chapter following introduction, background and methodologies.   The results are well presented follower by results dicutions and conclusions.

I suggest to complete the paper with following:

- Structure the contributions in comparation with another works in the field;

- To compare the results with another similar papers and highlighting the result but also the contributions.

Author Response

Dear reviewer,

Thanks for taking the time to read the manuscript and provide your constructive comments for improvements.

Here lists our response to all your comments:

  • Structure the contributions in comparation with another works in the field;

We have added subsection 6.2 Contributions from line 449 to structure our contributions. 

  • To compare the results with another similar papers and highlighting the result but also the contributions.

We have added Section 5.8, starting from line 410, to compare the accuracy of our proposed model with other published work that used the same dataset and found that our model has the best accuracy. Section 6 Discussion and Section 7 Conclusions and future work are rewritten to highlight our results and contributions. 

Thanks again for your time to read our revised version and this response letter to your comments.

Best Regards,

Yan Hu

Reviewer 4 Report

The manuscript proposes a technique for engagement recognition in the e-learning environment. The topic is very timely and significant for proper monitoring of remote education. Authors have used an optimized CNN model with lightweight architecture for the benefit of reduced calculation. They have designed the experiment in an organized manner but their literature survey, presentation of their analysis, and evaluation need more work. In my opinion, the manuscript is suitable for publication in the “Applied Sciences” journal only after resolving the major and minor issues given below.

Major points:

1.      Authors need to add a sentence in the abstract regarding why they have used ShuffleNet v2.

2.      The authors did not provide any reference related to existing works from 2022 and 2021 (only one work from 2020). There have been a significant amount of work done with CNN, engagement recognition, and the E-Learning environment during those years. Authors must go through relevant articles in recent years and make a significant amount of revisions in the literature survey (section 3).  

3.      References should be included for sentences from lines 53-65. Unless authors have proved any of those sentences later, they need to provide references.

4.      Authors need to make sure they have permission for the figures they have used from other articles.

5.      (line 228), so do authors think that reduced calculation amount is the only reason for ShuffleNet to be used in their technique?

6.      Need to provide more details on the dataset which was used here. Such as the age range of students or which grade they were in or for how long the video was recorded at a time etc.

7.      This dataset has been used in other existing articles for e-learning environments, authors need to provide a table that compares the accuracy, F measures, precision, recall, and recognition speed of their technique with other recently published works/techniques.  

Minor points:

1.      “for instance, teachers cannot obtain the feedback of students’ engagement in real-time” (line 3), do the authors think that currently no technology available to get real-time feedback from students over distance education? (for example use of “Clickers”).

2.      Line 10: what kind of real-world applications?

3.      Line 45-53, needs to revise, authors should not directly copy sentences from somewhere else changing just one or two words in every sentence.

4.      Similar action needs to be taken (as pointed out in minor point number 3) for lines 74-80.

Author Response

Dear reviewer,

Thanks for taking the time to read the manuscript and provide your constructive comments for improvements.

Here lists our response to all your comments:

Major points:

  1. Authors need to add a sentence in the abstract regarding why they have used ShuffleNet v2.

Our response: The Abstract is rewritten, and the sentence is added in line 6-7.

2. The authors did not provide any reference related to existing works from 2022 and 2021 (only one work from 2020). There have been a significant amount of work done with CNN, engagement recognition, and the E-Learning environment during those years. Authors must go through relevant articles in recent years and make a significant amount of revisions in the literature survey (section 3).  

Section 3 is extended by 8 references that were published in 2021 and 2022.

3. References should be included for sentences from lines 53-65. Unless authors have proved any of those sentences later, they need to provide references.

References [16][17] are addd.

4. Authors need to make sure they have permission for the figures they have used from other articles.

We got permission to use them by citing the articles. 

5. (line 228), so do authors think that reduced calculation amount is the only reason for ShuffleNet to be used in their technique?

No, we have added now in line 256-259 "Among different lightweight CNN models, ShuffleNet v2 has also verified its better performance on many direct metrics, such as speed, latency, and memory access cost, rather than only considering the approximate factor like float-point operations(FLOPs)."

6. Need to provide more details on the dataset which was used here. Such as the age range of students or which grade they were in or for how long the video was recorded at a time etc.

All the information on the dataset now is added in Section 4.3, starting from line 293.

7. This dataset has been used in other existing articles for e-learning environments, authors need to provide a table that compares the accuracy, F measures, precision, recall, and recognition speed of their technique with other recently published works/techniques.  

Section 5.8 (starting from line 410) is added to compare the accuracy of the proposed model with other existing models or methods, the result showed that our proposed model outperforms all other models in accuracy.  However, F measures, precision, recall, and recognition speed of their techniques were mostly not reported. 

Minor points:

1. “for instance, teachers cannot obtain the feedback of students’ engagement in real-time” (line 3), do the authors think that currently no technology available to get real-time feedback from students over distance education? (for example use of “Clickers”).

The original sentence causes some confusion. We have changed it to "For the teachers, it is more challenging to know the students learning efficiency and engagement when giving lectures online. " (line 3 to 4) for better understanding.

2. Line 10: what kind of real-world applications?

The original sentence causes some confusion. We have removed real-wrold applications and changed it to "In addition, this model is suitable for students' engagement recognition in distance learning on mobile platforms." (line 12 to 13) for better understanding.

3. Line 45-53, needs to revise, authors should not directly copy sentences from somewhere else changing just one or two words in every sentence.

These sentences are rewritten by our own words now (line 47-54).

4. Similar action needs to be taken (as pointed out in minor point number 3) for lines 74-80.

These sentences are rewritten by our own words now (line 86-93).

Thanks again for your time to read our revised version and this response letter to your comments.

Best Regards,

Yan Hu

Round 2

Reviewer 1 Report

The authors took into account the reviewers' recommendations - they extended the literature list, added RQs and commented them into the study, the extended the conclusion. I do not have any comments and notes anymore. 

Author Response

Thanks very much for reading the revision. We're glad that you are satisfied with our improvement.

Reviewer 2 Report

The authors partially answered some of the issues raised in the first review. The writing quality improved somewhat, though the article still needs careful editing. Results are now presented for several other approaches, which helps validate the importance of this work. The research questions of the study and relevance of the contributions achieved are now clearly presented. I think the article can be considered for publication, but it still needs to be carefully edited.

Author Response

Thanks very much for your suggestion. We have asked a professional native English speakers team to edit the manuscript, so we believe the paper's language quality is now sufficient to publish.

Reviewer 4 Report

The authors have answered the queries satisfactorily. 

Author Response

(The authors gave the same response as above.)
